# Elucidating the Local Transmission Dynamics of Highly Pathogenic Avian Influenza H5N6 in the Republic of Korea by Integrating Phylogenetic Information

**DOI:** 10.3390/pathogens10060691

**Published:** 2021-06-02

**Authors:** Dae-Sung Yoo, Byungchul Chun, Kyung-Duk Min, Jun-Sik Lim, Oun-Kyoung Moon, Kwang-Nyeong Lee

**Affiliations:** 1Department of Public Health, Korea University Graduate School, Seoul 02841, Korea; chun@korea.ac.kr; 2Department of Preventive Medicine, Korea University College of Medicine, Seoul 02841, Korea; 3Institute of Health and Environment, Graduate School of Public Health, Seoul National University, Seoul 08826, Korea; fortop@snu.ac.kr; 4Department of Public Health Sciences, Graduate School of Public Health, Seoul National University, Seoul 08826, Korea; borizook@snu.ac.kr; 5Import Risk Assessment Division, Animal and Plant Quarantine Agency, Gimcheon 39660, Korea; vetmoonok@korea.kr; 6Avian Influenza Research and Diagnostic Division, Animal and Plant Quarantine Agency, Gimcheon 39660, Korea; leekwn@korea.kr

**Keywords:** avian influenza, controlling strategy, culling, HPAI, H5N6 subtype, local transmission, spatial analyses, phylogenetic, poultry, transmission kernel

## Abstract

Highly pathogenic avian influenza (HPAI) virus is one of the most virulent and infectious pathogens of poultry. As a response to HPAI epidemics, veterinary authorities implement preemptive depopulation as a controlling strategy. However, mass culling within a uniform radius of the infection site can result in unnecessary depopulation. Therefore, it is useful to quantify the transmission distance from infected premises (IPs) before determining the optimal area for preemptive depopulation. Accordingly, we analyzed the transmission risk within spatiotemporal clusters of IPs using transmission kernel estimates derived from phylogenetic clustering information on 311 HPAI H5N6 IPs identified during the 2016–2017 epidemic, Republic of Korea. Subsequently, we explored the impact of varying the culling radius on the local transmission of HPAI given the transmission risk estimates. The domestic duck farm density was positively associated with higher transmissibility. Ring culling over a radius of 3 km may be effective for areas with high dense duck holdings, but this approach does not appear to significantly reduce the risk for local transmission in areas with chicken farms. This study provides the first estimation of the local transmission dynamics of HPAI in the Republic of Korea as well as insight into determining an effective ring culling radius.

## 1. Introduction

Highly pathogenic avian influenza (HPAI) virus (HPAIv), a member of influenza virus type A, is an infectious pathogen of poultry and wild birds. In particular, as indicated by its name, the virus causes high rates of mortality in infected chickens. With the viral genome consisting of eight RNA segments, novel, genetically different viruses are constantly evolving, which poses a challenge to domestic poultry industries worldwide [1].

Since HPAIv infection was first reported at a chicken farm in the Republic of Korea (ROK) in 2003, ring culling at HPAI-affected farms and geographically neighboring farms, such as within a 3 km radius, has been commonly implemented as a controlling strategy. However, strategies employing massive preemptive depopulation are severely damaging to poultry production systems and animal welfare. According to a report on the HPAI epidemic issued by the Korea Rural Economic Institute, 1.08 trillion Korean won (equivalent to approximately 980 million US dollars) have been spent on direct compensation payments for culling. Approximately 94.15 million birds have been culled as of 2018, and long-term maintenance and monitoring to detect environmental contamination thereafter are necessary [2].

In principle, the current culling radius in the ROK is determined based on local spread between infected premises (IPs) and susceptible ones. Local spread refers to local transmission (within 3 km) from an unknown infection source or via an unknown infection pathway [3]. However, it could be argued that the inter-farm transmission distance can vary with the degree of pathogen infectivity, farm density, and geographical conditions related to virus dispersal and survival. Moreover, a recent study employing mathematical modeling reported that the inter-farm transmission distance during the HPAI epidemic in the ROK depended on the density of the farms [4]. The paper also recommended that the current preemptive culling distance should be adjusted to minimize economic losses and adverse effects. Hence, quantification and observation of the transmission distances between IPs should be performed as part of the process to establish the optimal perimeter for preemptive depopulation.

Over the last decade, many epidemiological studies have attempted to estimate the average HPAI transmission distance between infected farms using transmission kernels. The transmission kernel is a geographical indicator defined as the probability distribution function of the distance between sequential cases in a transmission chain, and it is generally used to determine the minimum radii of protection and surveillance zones [5]. The transmission kernel can be used to understand the spatial range over which a contagious pathogen is transmitted between susceptible premises and IPs [6].

The use of transmission kernels to estimate inter-farm transmission distances has several advantages when planning and implementing preparedness and emergency responses to avian influenza epidemics, including preemptive depopulation. For example, one study showed that, in The Netherlands, an intensive depopulation policy appeared to be the most effective control measure against HPAI outbreaks in poultry-dense regions where geographically proximal infections were likely to occur [7]. Similarly, a simulation study based on transmission kernel estimates maintained that preemptive culling at an earlier phase of the epidemic can minimize economic losses from HPAI outbreaks in Italy because of the short transmission distance [8]. Furthermore, recent studies have introduced the potential application of HPAI-risk maps constructed using basic reproduction numbers, which are estimated from the transmission kernel function [9,10]. These works advised that it is necessary to measure inter-farm transmission distances when designing intervention strategies to lower financial costs and obtain optimal epidemic outcomes.

Despite the usefulness of this indicator, it is also associated with a significant limitation. When calculating the transmission kernel, it is presumed that inter-farm transmission takes place among all IPs. In practice, this simplified assumption is likely not valid because there are remarkable phylogenetic differences among viruses sampled from different IPs. This means that some farms are probably not directly linked with other infected farms in terms of virus transmission; thus, in the estimation of transmission kernels, non-linked farms have to be excluded from the analyses [11]. Accordingly, for the robust estimation of transmission parameters, phylogenetic information for all IPs within the area of interest must be incorporated, even if there is uncertainty associated with the inference of transmission linkage [12]. Nevertheless, very few attempts to measure the magnitude of inter-farm transmission using an integrative approach have been made [13], despite the growing need for analyses to include a combination of epidemiological and genetic data.

In this regard, we developed a novel approach for enhancing the reliability of estimating spatial transmission kernel parameters for the HPAI H5N6 epidemic from 2016 to 2017 in the ROK by integrating phylogenetic and epidemiological information. First, we identified spatiotemporal clusters where IPs with viral strains in the same phylogenetic group were locally concentrated. Next, given the clustering area, the transmission kernel was estimated, assuming that the virus was only transmitted between IPs in the same phylogenetic cluster. Finally, we ran simulations with the kernel estimates to assess the impact of different preemptive culling strategies on local transmission during the 2016 to 2017 HPAI epidemic. This study provides insight into the local transmission dynamics of HPAI H5N6, and the results will help decision-makers devise effective control strategies over the course of epidemics that may emerge in the future.

## 2. Materials and Methods

### 2.1. Data Source

Data on the HPAI outbreak and poultry holdings subjected to spatiotemporal cluster analyses and spatial transmission kernel estimates were obtained from the ROK Animal and Plant Quarantine Agency (https://home.kahis.go.kr/home/lkntscrinfo/selectLkntsOccrrncList.do (accessed on 20 May 2019)). According to the 2018 Terrestrial Manual of the World Animal Health Organization [14], a positive result for HPAI virus from a real-time reverse transcription-quantitative polymerase chain reaction (RT-qPCR) test defines an HPAI outbreak at a poultry holding. During the HPAI H5N6 epidemic, veterinary authorities implemented a proactive surveillance program, in which all duck farms had to be tested for HPAI infection and chicken farms were inspected every week [15]. The enhanced surveillance was enforced from 28 October 2016 (the date on which HPAIv was first detected in wild birds) until the end of July 2017.

There were 343 HPAI IP and 7954 non-IP across Korea during the 2016–2017 HPAI H5N6 epidemic. Of the IP, 57.4% were chicken farms, 40.2% were duck farms, 2.0% were quail farms, and 0.4% were farms for other species. Moreover, 72.0% were located in central Korea, with the rest being in the southwest and north. Data included geographical coordinates, flock sizes (mean = 69,071 birds; range: 13–787,533 birds), dates of HPAIv detection/reportage thereof (between 16 November 2016 and 3 March 2017), the start date of culling for IPs (between 16 November 2016 and 4 March 2017) and start dates of pre-emptive culling for non-IP if applicable. In ROK, the poultry premises were mainly chicken and domestic duck farms, which accounted for 85.4% and 14.1% of all poultry holdings, respectively, as of July 2016. Chicken farms are evenly distributed across Korea, while duck farms are localized in the central Eumseong and southwestern Naju regions (Appendix A).

Phylogenetic clustering of the IP was performed in previous studies, from which cluster numbers (i.e., clusters 1–5 [C1–5]), were extracted [7,15,16,17]. For the phylogenetic analyses in a previous study [16], RNA extracted from HPAI-positive poultry was matched with nucleotide sequences downloaded from Global Initiative for Sharing All Influenza Data (GISAID) (http://www.gisaid.org (accessed on between 16 November 2016 and 4 March 2017)) using CLC workbench software (ver. 6.8.2; CLC Bio, Aarhus, Denmark). Based on the complete nucleotide sequences of hemagglutinin (HA), matrix (M), neuraminidase (NA), nucleoprotein (NP), nonstructural (NS), polymerase acidic (PA), polymerase basic 1 (PB1) and polymerase basic 2 (PB2), maximum likelihood (ML) trees were constructed using nucleotide substitution models (Hasegawa–Kishino–Yano model with a gamma (γ)-distribution for HA and NP; general time reversible model with a γ-distribution for PB2, PB1, PA, and NA; Kimura 2-parameter model with a γ-distribution for M; Tamura 3-parameter model with a γ-distribution for NS) and MEGA 6 software (www.megasoftware.net (accessed on between 16 November 2016 and 31 December 2019)). The H5N6 viruses isolated from poultry farms were classified into four distinct clusters (C2–C5) based on the homologies of the PA and NS genes.

### 2.2. Spatiotemporal Cluster Analyses

We first conducted cluster analyses to identify the areas where IPs with the same types of HPAI H5N6 virus were geographically and temporally clustered. The assumption was that local transmission was more likely to happen among clustered farms, such that the probability of remote viral transmission was minimized. The phylogenetic cluster numbers assigned to the IP were included as categorical variables (C2 = 1, C3 = 2, C4 = 3, C5 = 4, no genotype = 5) in a statistical analysis of spatiotemporal heterogeneity. The multinomial distribution allowed us to distinguish the IP genotypes. In the analysis, relative risk corresponded to the ratio between the observed and expected number of cases in each cluster. The multinomial scan window searches for a cluster where the number of infections due to each genotype is statistically higher than that outside the cluster [18]. The null hypothesis (*H*_0_) was that there was no clustering and the occurrence probability of a HPAI H5N6 genotype *k* was the same for *k* = 1, 2, 3, 4, 5 in all study areas. The alternative hypothesis was that there was at least one H5N6 genotype for which the occurrence probability within a cluster was higher than that outside the cluster.
(1)$$H_{0}:p_{1}=q_{1},\ldots \ldots p_{k}=q_{k},H_{1}:at\;least\;one\;p_{k}>q_{k}$$
where *p_k_* and *q_k_* are the probabilities of genotype *k* being located inside the scanning window Z and outside the window, respectively. The likelihood function for the multinomial model is then expressed as follows:(2)$$L\left(Z\vee p1,\ldots p5,q1,\ldots q5\right)\propto \overset{5}{\underset{k=1}{\prod }}\left(\underset{i\in Z}{\prod }p_{k}^{c_{ik}}\underset{i\notin Z}{\prod }q_{k}^{c_{ik}}\right),L_{0}=\underset{k}{\prod }{\left(\frac{C_{k}}{C}\right)}^{C_{k}}$$
where *c_ik_* is the number of HPAI H5N6 IPs linked to genotype *k* in region *i*, *c_i_* is the total number of HPAI H5N6 cases in region *i*, *C_k_* denotes the total number of HPAI H5N6 cases with genotype *k*, and *C* is the total number of observations in the whole study area. *L*_0_ is a multinomial likelihood function under the null hypothesis. The most likely spatiotemporal cluster is the one that maximizes the likelihood ratio test statistic. Under the null hypothesis, the likelihood ratio test statistic is written as:(3)$$Likelihood\;ratio=\frac{L\left(Z\right)}{L_{0}}$$

The significance of the likelihood ratio for candidate cluster *Z* was estimated using a Monte-Carlo simulation. The likelihood ratio was calculated based on the scanning window’s circular shape. We set the maximum cluster size to 20% of the population at risk to identify regions likely to experience local transmission. This threshold provided the optimal outcomes in a previous study conducted in the ROK [19]. In addition, The time period was set to 21 days, which corresponds to the maximum incubation period between 27 October 2016 and 3 March 2017. [20]. Significant clusters were limited to areas with more than 30 cases each to maintain the statistical power for calculating transmission kernel estimates. SatScan version 9.7 was used to perform cluster analyses [21].

### 2.3. Estimation of Spatial Transmission Risk

We hypothesized that local spatial transmission would occur between poultry farms within a spatiotemporal cluster with a high rate of HPAI infections. To test this hypothesis, we derived the force of infection based on a transmission kernel function, in accordance with previous research [8,10,22], as follows:(4)$$h\left(d_{ij}\right)=\frac{h_{0}}{\left(1+{\left(d_{ij}/r_{0}\right)}^{\alpha }\right)}$$
where *d_ij_* represents the Euclidean distance between HPAI-infectious farm *i* and susceptible farm *j*, h_0_ refers to the maximum hazard rate at *d_ij_* = 0, and *r*_0_ is the half-value distance parameter that determines the distance influenced by the hazard rate. The parameter α is related to the decay rate of the hazard rate from its maximum value.

With respect to transmission kernel modeling, the HPAI infection status of individual farms were classified as susceptible, newly infected, infectious, or removed during each period of spatiotemporal cluster formation. As a result, the presence or absence of antibodies was not confirmed. Thus, we assumed that the infectious period lasts 7 days, which correspond to the duration between the start date of the infectious status and the reporting or sampling date [20]. An HPAI-infected farm is supposed to be infectious at 1 day after the infection [8,10,22]. An IP was also classified as having a “removed” status if culling started. Sensitivity analyses for different infectious intervals (14 vs. 21 days) were performed to choose the optimal combination of assumptions based on the Akaike information criterion (AIC). The infection status was updated daily.

To estimate the transmission risk between premises given infections by viruses with the same genotype in the area, the force of infection for farm *i* on day *t*, was calculated using a predefined transmission kernel function as the sum of hazard rates associated with various infectious farms *j*, as follows:(5)$$\lambda_{i\left(t\right)}=\underset{j\neq i}{\sum }h\left(d_{ij}\right)I\left[jisspecificgenotypeinfectious\right]$$
where *I* is an indicator function where if farm *j* is infectious *I* = 1, otherwise, *I* = 0. The probability of premises *i* remaining uninfected with a specific HPAIv genotype up to day *t*, *r_i_*(*t*), is calculated via the accumulative multiplication of the probability of being uninfected on each day until day *t*, as follows:(6)$$P\left(T\geq t\right)=r_{i}\left(t\right)=e^{-\sum_{s=1}^{t-1}\lambda_{i}\left(s\right)}$$

For the efficient inference of parameters, the premises were reclassified into four compartments based on infection status and phylogenetic similarity, as described in Appendix A). With K the group of uninfected farms that were not depopulated during the epidemic, let F be the set of farms that was uninfected and underwent preemptive culling on day *t_preculled,f_*, M be the fraction of premises infected with the virus genotype of interest on day *t_inf,m_.*, with these premises considered infectious farms, and D be the group of infected farms with other HPAIv genotypes that were considered uninfected and underwent culling on day *t_culled,d_* (see Appendix A). Under the assumption that the event at each time t is an independent occasion, the probability of infection at each point is multiplied over the entire epidemic duration. the log-likelihood (ı) can be expressed in terms of the force of infection and using the equations for *q_i_*(*t*) and *r_i_*(*t*), as follows:(7)$$\begin{array}{cc} ı=-\underset{k\in K}{\sum }\overset{t_{max}-1}{\underset{t=1}{\sum }}\lambda_{k} & \left(t\right)\\ & -\underset{f\in F}{\sum }\overset{t_{preculled}-1}{\underset{t=1}{\sum }}\lambda_{f}\left(t\right)\\ & -\underset{d\in D}{\sum }\overset{t_{culled}-1}{\underset{t=1}{\sum }}\lambda_{d}\left(t\right)\\ & -\underset{m\in M}{\sum }\overset{t_{infected}-1}{\underset{t=1}{\sum }}\lambda_{m}\left(t\right)+\underset{m\in M}{\sum }log\left(1-e^{-\lambda_{m}\left(t_{infected}\right)}\right) \end{array}$$
where *t_max_* is the final day of the outbreak in a given spatiotemporal cluster. In this analysis, we excluded the farms that stopped production or carried out culling before the start date of spatiotemporal cluster formation. With this log-likelihood formula, the transmission kernel parameter was estimated using maximum likelihood estimation with the limited-memory Broyden–Fletcher–Goldfarb–Shanno optimization algorithm [9]. The R bbmle package was used to perform maximum likelihood estimation [23].

### 2.4. Identification of Factors Associated with the Local Transmissibility of Highly Pathogenic Avian Influenza (HPAI)

The basic reproduction number, which represents the number of secondary cases caused by one infectious host and reflects disease transmissibility, was estimated for each IP in two spatiotemporal clusters, to determine the transmissibility for individual IP based on the spatial transmission kernel *h*(*d_ij_*) and stochastic infectious period (*T_i_*), which ranged from 1 to 7 days. The reproductive number of IP *i*, *R_IP_,_i_* was calculated following previous research [6] (Appendix A):(8)$$R_{IP,i}=\underset{i\neq j}{\sum }\left(1-E\left[e^{h\left(d_{ij}\right)\times T_{i}}\right]\right)$$

To identify factors associated with the basic reproduction number of the IP, five variables were considered for each IP: flock size, the duck and chicken farm population densities, the human population density, and the minimum distance to a driveway from the IP. The density of duck and chicken farm populations was estimated for individual IP using a kernel density function, where the bandwidth of the kernel was set to 3 km and density was determined at 100 m cell size (Appendix A). The human population density was estimated by averaging values over 1-km cells, using 2016 data from WorldPop (https://www.worldpop.org/ (accessed on 25 January 2020)). The minimum distance was calculated as a Euclidean distance, using 2016 data from the Korean National Transport Information Center (https://www.its.go.kr/nodelink/intro (accessed on 25 January 2020)). Finally, a mixed linear regression model was built to assess the associations between the basic reproduction number and five explanatory variables using the lmerTest package (ver. 3.1.3) [24].

### 2.5. Simulation of Ring Culling Radii

Using simulations, we examined the potential impact of varying the culling radius (0.5, 1, 2, or 3 km from an IP) on the epidemic. To compare the outcome through a simulation from the practical point of view, we selected those four different culling radii that were used as a controlling measure against HPAI outbreaks according to the past and present protocol. The depopulation radius was 500 m before 2017 but was extended to 3 km after the massive 2016 to 2017 HPAI H5N6 outbreak. During the 2020 to 2021 HPAI H5N8 epidemic, the culling radius was adjusted to 1 km from an IP. Each simulation included one observed IP randomly selected as an index farm assumed to be at the latent stage (newly infected) of infection at the beginning of the simulation, and HPAI infections only occurred with a threshold force of infection based on given transmission kernel estimates. Simulations were performed for all outbreaks found in the clusters. In terms of epidemic outcomes, we compared the number of IPs and the number of farms that underwent preemptive culling (PCs). The median and 5th to 95th percentile (5–95 PCTL) range of those outcomes were calculated by generating 1000 iterations of each scenario with a discrete time step of 1 day over the duration of the 2016–2017 HPAI epidemic. The simulations were performed using R software version 4.0.4.

## 3. Results

### 3.1. Spatiotemporal Characteristics of the HPAI H5N6 Epidemic

As described in Table 1, a total of 343 poultry premises were infected with HPAIv for 108 days of the 2016 to 2017 HPAI H5N6 epidemic in the ROK. On average, each IP was located approximately 66.71 km (95% confidence interval [CI] 0.15–272.43) from another IP Infections with the C4 genotype (134 IPs) lasted 86 days, with this genotype accounting for the highest case number among the four genotypes. For infections with this genotype, the mean distance between IPs was 75.72 km (95% CI 0.10–283.46). By contrast, 58 poultry holdings were reported as C3 infection cases over 51 days, and on average, they were located 29.36 km (95% CI 0.10–283.46) from one another. Infections with the C2 genotype were confirmed on 97 poultry premises over 64 days, whereas infections with the C5 genotype were reported on 33 farms over 45 days. For both the C2 and C5 genotypes, the infection cases occurred within a mean of 50.93 km from one another (Figure 1).

### 3.2. Cluster Analyses

According to the spatiotemporal cluster analyses based on multinomial scan statistics, two clusters of poultry premises were identified as being at significantly high relative risk (RR) of genotype-specific H5N6 outbreaks (Figure 2). The two spatiotemporal clusters differed in size (cluster A, 36.1 km^2^; cluster B, 8.32 km^2^) but were geographically conjoined at the center of the country during an overlapping period from mid-November 2016 to mid-January 2017. In cluster A, C3-genotype H5N6 viral infections were predominant relative to those of other HPAIv genotypes (RR 41.28) from 26 November 2016 to 6 January 2017 (Table 2). Of 58 HPAI cases, 42 were confirmed as infections with the C3-genotype virus. In cluster B, the IPs had a highly localized distribution, with only C4-genotype infections observed (RR 3.43) within an area with a radius of 8.32 km. In this area, HPAI outbreaks were reported on 48 farms from 16 November 2016 to 16 December 2016.

### 3.3. Spatial Transmission Kernel Estimates

Table 3 summarizes the transmission kernel estimates related to the probability of a susceptible farm being infected by an IP located a variable distance from it. There was a pronounced disparity in the maximum hazard rate (h_0_) estimated from the outbreaks in two clusters. The transmission kernel estimates of cluster A, which lay within a circular area with a radius of 36.1 km, produced lower infection hazard rates, with a mean of 0.62 × 10^−3^ (95% CI 0.42 × 10^−3^–0.82 × 10^−3^) at zero distance. By contrast, the maximum hazard rate according to the transmission kernel estimates for cluster B, which lay within a circular area with a radius of 8.2 km, was higher, with a mean value of 2.62 × 10^−3^ (95% CI 1.61 × 10^−3^–3.63 × 10^−3^). For cluster A, estimates of r_0_ (mean 2.603, 95% CI 2.350–2.857), which is the distance at which the hazard rate is half its maximal value, were comparable to those for cluster B (mean 2.246, 95% CI 0.555–3.936). Similarly, there was no obvious difference in the decay rate (α) of the daily infection hazard posed by IPs between the two clusters, with a mean rate of 1.363 (95% CI 0.928–1.797) for cluster A and a mean rate of 1.358 (95% CI 0.351–2.365) for cluster B. Given these transmission kernel estimates and an assumed infectious period of 7 days, the daily probability of HPAI infection of a susceptible farm via transmission from IPs with infectious HPAIv located 0.5 and 10 km away was estimated to decrease slightly from 0.004 to 0.001 in cluster A, whereas the probability fell sharply from 0.016 to 0.002 in cluster B (Figure 3). Sensitivity analysis exhibited the infectious duration of seven days lowest AIC value both cluster A and cluster B (Appendix A).

### 3.4. Factors Associated with the Local Transmissibility of HPAI

Table 4 summarizes the results of the mixed linear regression model of the basic reproduction number of the IP. The model included five explanatory variables related to environmental conditions and production size. Two variables had significant relationships with the lateral transmissibility of HPAI. The density of duck farms around IP had a positive association with the basic reproduction number (a mean of regression coefficient estimates [mean] = 0.237, 95% confidence interval [CI]: 0.177–0.306), while the density of chicken farms was inversely associated with the basic reproduction number (mean = −0.146, 95% CI −0.300 to −0.003). The flock size of IP (mean = −0.004, 95% CI −0.011 to 0.004), human population density (mean = 0.001, 95% CI −0.016 to 0.017), and minimum distance to a driveway from an IP (mean = −0.156, 95% CI −0.479 to 0.174) were not significantly associated with the basic reproduction number.

### 3.5. Simulations with Different Preemptive Culling Radii

Figure 4 illustrates the cumulative median number IPs and PCs generated from simulations based on four culling radii (i.e., 0.5, 1, 2, and 3 km) of clusters A and B. Additionally, Appendix A enumerates 5th to 95th percentile (5–95 PCTL) range of cumulative number IPs and PCs during the average extinction days of 1000 iterations. The longer the culling radius, the greater the number of PCs and the smaller the number of IPs. However, the variance in the numbers of IPs and PCs across the culling range differed between clusters A and B. In cluster A, the number of IPs did not change much as the culling radius increased (0 to 1 IP). For example, with a culling radius of 0.5 km, 18 premises (5–95 PCTL 8–31) were affected by HPAI and 578 farms (5–95 PCTL 365.95–646) had to implement culling over 32 days on average, whereas with a radius of 3 km, a mean of 17 premises (5–95 PCTL 8–30) were affected by HPAI and a mean of 609 farms (5–95 PCTL 534.95–636) had to implement culling over 28 days.

By contrast, the numbers of IPs and PCs differed over a broader range between culling radii of 0.5 and 3 km in the simulation of cluster B (mean difference [MD] 12 IPs and 15 PCs). However, no obvious differences were observed for the numbers of IPs and PCs between culling radii of 2 and 3 km (MD 2 IP and 3 PCs). There was a distinctive difference in the epidemic outcome from simulation on between a culling radius of 0.5 km and 3 km. With a culling radius of 0.5 km, a mean of 67 poultry holdings (5–95 PCTL 41–92.025) were infected with HPAIv and a mean of 103 farms (5–95 PCTL 80–125) implemented culling over 32 days, whereas with a radius of 3 km, 52 IPs (5–95 PCTL 29–77.025) and 118 PCs (5–95 PCTL 94.975–136) over 30 days were projected on average.

## 4. Discussion

This study evaluated the relative infection risk posed by lateral transmission between farms located at variable distances rather than other types of transmission pathways, including long-range viral transfer by infected poultry or vehicle movements between farms. To accomplish the primary objective of the study, we identified spatial and temporal concentrations of HPAI occurrences among poultry holdings, where inter-farm transmission was more likely to occur. In addition, we assumed that direct transmission was unlikely between premises infected with phylogenetically different viral strains. Therefore, we estimated transmission risk based on the force of infection between premises infected with HPAIv with the same genotype, using phylogenetic cluster information extracted from previous studies [7,16,17]. This approach allowed us to calculate more reliable spatial transmission estimates. Moreover, we simulated different culling scenarios to examine how different control strategies affect the local transmission risk.

In multinomial spatiotemporal cluster analyses, two clusters were found to have a significantly higher probability of infection with specific genotypes of the H5N6 virus compared to other clusters. These two regions with a concentration of HPAI infections developed at the early phase of the 2016 to 2017 HPAI H5N6 epidemic, with the infections progressing almost concurrently during the epidemic. Concurrent but separate outbreaks arising from the same genotype indicate that local transmission among poultry holdings was prevalent, because the viral genotypes associated with IPs would be randomly distributed if other viral strains were frequently introduced. Specifically, in cluster B, 48 cases involving the same HPAIv genotype were reported over an approximate period of 1 month. Considering the incubation period and reporting intervals from the infection dates, it is likely that the actual infections started before the intensive interventions implemented by the veterinary health authority. Because the implementation of additional on-farm preventive measures against a potential infection source is unlikely, usual farming practices such as transporting feed and eggs probably accelerate transmission upon the introduction of a pathogen. Under these circumstances, widespread local transmission is very likely. Therefore, it is critical to quantify spatial transmission risk.

In the transmission kernel analysis, the infection hazard posed by an HPAI infectious premise at zero distance from a susceptible farm showed a distinctive difference. The daily maximum hazard rate in cluster B was approximately 4.2 times higher than that (0.57 × 10^−4^) in cluster A. This variation could be explained by the result of the association test between Ips’ basic reproduction number and geographical characteristics. As given in Table 4, the higher density of domestic duck farms around IPs contributed to increasing the lateral transmissibility of HPAIv. Previous studies have reported that density duck farms were found one of the significant risk factors for HPAI infection [25,26]. In fact, HPAI infected domestic ducks do not generally show distinct symptoms that farmworkers can readily recognize and place control action. Also, the domestic duck holdings generally had a low biosecurity protocol level in operation, resulting in inadequate protection against viral incursion and prolonged period time for disseminating virus [20]. Moreover, most HPAI infected duck holdings in those regions were operated by a couple of poultry integrators (not shown here), driving highly close connections between farms by sharing the same production resources such as feed suppliers.

By contrast, domestic chicken farm density showed a negative relationship with lateral transmissibility of HPAIv. Infected chicken usually presented noticeable clinical signs, including death, dropped egg production rate, and increased daily fatality rate leading to early detection to prevent further spread. Additionally, a high biosecurity compliance protocol is exacted in chicken farms compared to domestic duck farms. Collectively, geographical and epidemiological characteristics are associated with the probability of pathogen propagation among neighboring farms in the cluster [27,28].

The transmission kernel in this study displayed a similar configuration as that reported in previous studies that used the same transmission kernel function. The transmission kernel estimates for outbreaks in The Netherlands during the 2003 HPAI H7N7 epidemic were similar to those for cluster B, with a maximum hazard rate of 2.0 × 10^−3^ per day and a decay rate of 2.1 [29]. During the epidemic, outbreaks originated in high-density poultry production regions in the Netherlands, where the virus initially spread rapidly to neighboring farms before stringent control measures including preemptive depopulation and movement restrictions were enacted [30,31]. In addition, parameter estimates for the outbreaks that occurred during the 1999–2000 HPAI H7N1 epidemic in Italy [8], such as the spatial range of influence (r_o_), were similar to those estimated for the outbreaks in our study. A substantial number of domestic duck holdings were affected in Italy, and afterward, a strict contingency strategy was also enforced [32]. Due to similarities in demographic conditions and the intensity of the response measures, we found comparable transmission kernel estimates, maximum hazard rates, and risk reduction (all rapid) between these studies and ours.

Based on the spatial transmission kernel estimates, the simulation model indicated that a 2 or 3 km culling radius appeared to be more appropriate for high-density duck farm regions than a radius of less than 0.5 km or 1 km. This is consistent with the optimal culling radius (2.24 km) determined by a previous mathematical simulation study of a high-density duck farm region in the ROK that would minimize the number of premises that would need to implement culling and maximize the number of non-infected premises [4]. This suggests that if the large-scale culling in cluster B farms were not implemented, the hazard risk faced by a susceptible farm (i.e., transmission from infectious premises) would be higher than that estimated in the present study. Thus, a 3 km culling radius can reduce disease transmission among domestic duck farms in high-density regions, despite greater losses of poultry holdings.

Nevertheless, one can also argue that distance-based culling causes excessive loss of holdings, particularly if enhanced biosecurity and other production measures on the premises can sufficiently restrict viral propagation. In our simulations, increasing the culling range from 0.5 to 3 km did not considerably reduce the number of IPs in cluster A. The number of cases only decrease by one, whereas an additional 31 farms were depopulated. Another simulation study based on the transmission kernel function suggested a non-depopulation strategy for low-risk regions [22]. It appears that a standard culling radius is not optimal for poultry premises across the whole country because of variation in the transmission rate and vulnerability to infection. Therefore, culling radii should be determined based on the geographical conditions and biosecurity levels related to inter-farm transmission in outbreak regions.

We note here some of the more important limitations of this study. The spatial transmission kernel is an epidemiological metric based on the likelihood of pathogen transmission from infected to uninfected farms as a function of inter-farm distance. Hence, during parameter estimation, there was no consideration of long-range transmission from outside the defined clusters, such as from vehicle movements or wild birds. However, it is worth calculating the force of infection while accounting for local transmission, remote transmission, and wild bird-mediated infection to obtain better estimates of transmission distance. Furthermore, as mentioned above, because spatial transmission kernel estimates are highly influenced by diverse factors, including geographical conditions, host susceptibility, and on-farm biosecurity level, an in-depth study on the associations between transmission range and those factors is necessary. Regarding the simulation model, it is not ideal to use transmission kernel estimates as the input data, because the control measures implemented have already constrained the values. For example, r_o_ in cluster B might be much smaller than that under no control measures. Moreover, even though we simulated the impact of different culling radius on the HPAI outbreaks at poultry holdings in two spatiotemporal clusters, it did not provide the statistical significance to consolidate the difference of epidemiological outcomes resulting from adjusting the culling radius. Also, the economic costs and benefits of various culling radii which is worthwhile to understand the practical impact of chaining the culling radii in future work. Nonetheless, the simulations allowed a comparison of different intervention strategies under the same pre-conditions.

To the best of our knowledge, this study is the first to evaluate the transmission hazard range associated with HPAI and the probabilities of a susceptible farm being infected with HPAI from infectious premises located at different distances via the incorporation of molecular and epidemiological data. We inferred reliable estimates of transmission risk, which is fundamental to designing an effective control strategy. Based on our results, culling radii should be determined with consideration of the geographical conditions and farm species in the outbreak regions. The study provides novel insight into the optimal spatial range over which poultry holdings should implement contingency interventions to prevent the next HPAI epidemic.

## Figures and Tables

**Figure 1 pathogens-10-00691-f001:**
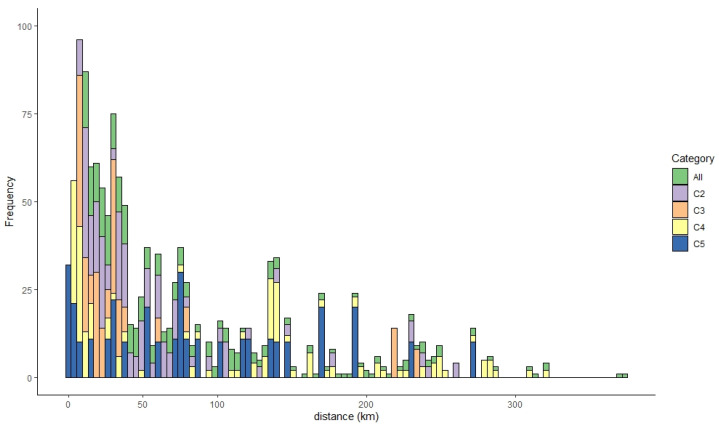
Distribution of distances between HPAI-infected premises for all virus genotypes and specific genotypes. The frequency of each distance range for each genotype category is stacked within a column. Green indicates all infected farms, and purple, orange, yellow, and blue indicate premises infected with the C2, C3, C4, and C5 HPAI virus genotypes, respectively. HPAI, highly pathogenic avian influenza.

**Figure 2 pathogens-10-00691-f002:**
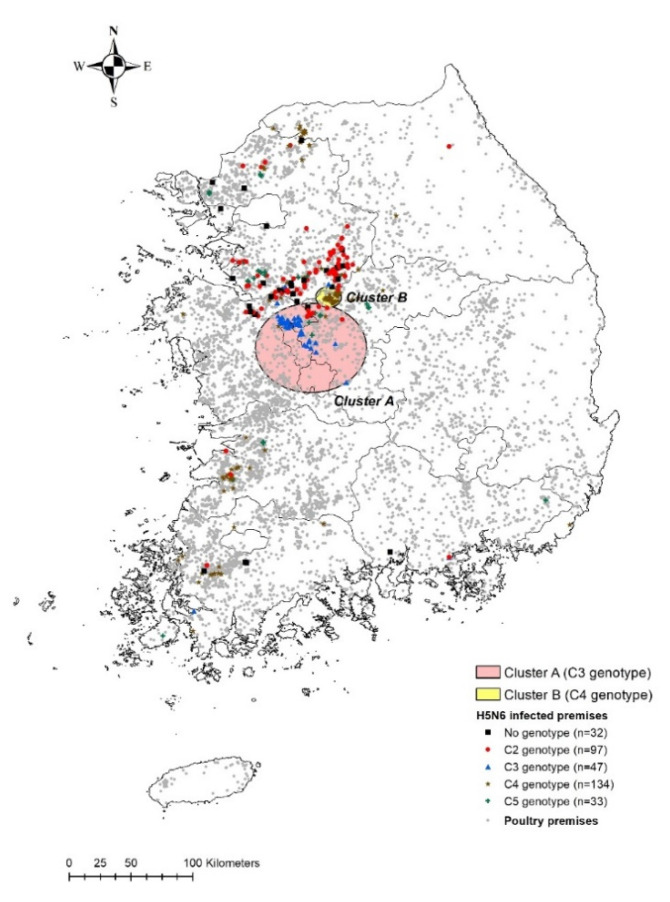
Spatiotemporal clusters of poultry farms involved in HPAI H5N6 outbreaks associated with four virus genotypes during the 2016 to 2017 epidemic in the Republic of Korea. Two significant clusters were identified. Farms in cluster A (red shading) were at higher relative risk (RR) of infection with the C3 genotype (RR 41.28), whereas those in cluster B (yellow shading) were at high risk for C4 infection (RR 3.43). Red dots, blue triangles, brown stars, and green crosses denote premises infected with the C2-genotype virus, C3 virus, C4 virus, and C5 virus, respectively. Gray dots indicate all other poultry holdings. HPAI, highly pathogenic avian influenza.

**Figure 3 pathogens-10-00691-f003:**
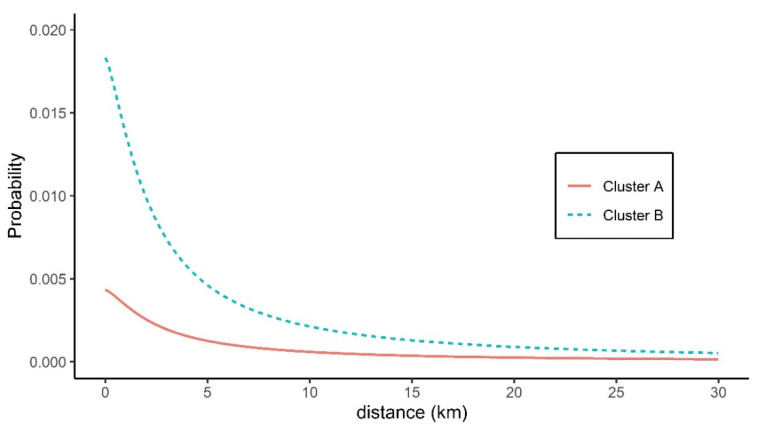
Daily probability of a susceptible farm in two clusters being infected by infectious premises according to inter-farm distance. The distance-based daily probability of HPAI infection at a poultry holding located at different distances from individual infectious premises was estimated using transmission kernel estimates for each cluster assuming an infectious period of 7 days. Spatial transmission kernel estimates were derived from clustered HPAI outbreaks associated with two specific virus genotypes. HPAI, highly pathogenic avian influenza.

**Figure 4 pathogens-10-00691-f004:**
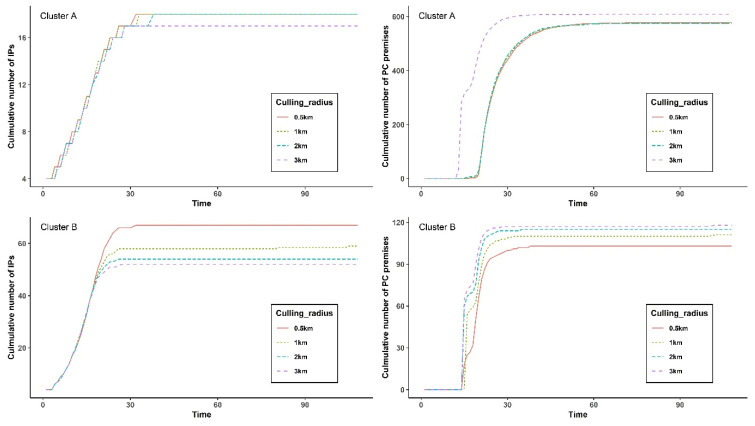
Cumulative number of infected premises (IPs) and preemptive culled premises (PCs) from simulation of four different pre-emptive culling ranges among the poultry holdings in cluster A (**top**) and cluster B (**bottom**); 0.5 km (red solid line), 1 km (green dotted line), 2 km (blue dot dash line) and 3 km (purple dashed line) from an IP. Top left and top right plots represented cumulative number of infected premises and pre-emptively culled premises in cluster A over the days. Bottom left and bottom right plots represented cumulative number of infected premises and pre-emptively culled premises in cluster B over the time (unit: days).

**Table 1 pathogens-10-00691-t001:** Spatial and temporal characteristics of the 2016 to 2017 highly pathogenic avian influenza (HPAI) H5N6 epidemic on poultry premises in the Republic of Korea (see Figure 1).

Genotype	Distance between IPs (km) Mean (95% CI)	Outbreak Duration (Start Date, End Date)	No. of Cases
All	66.71 (0.15, 272.43)	108 days (16 November 2016, 3 March 2017)	343
C2 genotype	50.93 (0.24, 234.88)	64 days (29 November 2016, 21 January 2017)	97
C3 genotype	29.36 (0.67, 218.29)	51 days (16 November 2016, 5 January 2017)	58
C4 genotype	75.72 (0.10, 283.46)	86 days (16 November 2016, 9 February 2017)	134
C5 genotype	50.93 (0.40, 240.37)	45 days (20 November 2016, 3 February 2017)	33

HPAI, highly pathogenic avian influenza; IPs, infected premises; CI, confidence interval.

**Table 2 pathogens-10-00691-t002:** Spatiotemporal clusters of poultry farms infected with four HPAI H5N6 virus genotypes in the Republic of Korea (see Figure 2).

Characteristics	Cluster A	Cluster B
Radius (km)	36.11	8.32
Duration (from start date to end date)	26 November 2016–6 January 2017	16 November 2016–16 December 2016
Total no. of IPs in the cluster	58	48
No. of IPs infected with the clustered genotype	42	48
RR associated with each genotype (C2, C3, C4, C5, N/A)	(0.44, 41.28, 0.074, 0.68, 0.33)	(0, 0, 3.43, 0, 0)
LLR	82.26	51.45
*p*-value	0.001	0.001

HPAI, highly pathogenic avian influenza; IPs, infected premises; RR, relative risk (ratio of the proportion of HPAI cases for each genotype out of the total number of HPAI cases within a cluster to that outside the cluster); LLR, log-likelihood ratio.

**Table 3 pathogens-10-00691-t003:** Spatial transmission kernel estimates for outbreaks in two clusters.

Cluster (Genotype)	*h* _0_	*r* _0_	α	AIC
Cluster A (C3)	0.00062 (0.00042, 0.00082)	2.603 (2.350, 2.857)	1.363 (0.928, 1.797)	633.99
Cluster B (C4)	0.00262 (0.00161, 0.00363)	2.246 (0.555, 3.936)	1.358 (0.351, 2.365)	483.64

Transmission kernel parameters are expressed as mean values with 95% confidence intervals in parentheses. *h*_0_ refers to the maximum hazard rate (zero distance between susceptible premises and infectious premises), *r*_0_ is the half-value distance of the maximum hazard rate, α is the decay rate of the hazard rate from its maximum value. AIC, Akaike information criterion; HPAI, highly pathogenic avian influenza; IP, infected premises.

**Table 4 pathogens-10-00691-t004:** Factors associated with the basic reproduction number of infected premises for HPAIv H5N6 outbreaks: results of a mixed linear regression model.

Variable (Unit)	Coefficient Estimates		*p*-Value
	Mean	95% CI Lower, Upper	
Flock size (10,000 head)	−0.004	−0.011, 0.004	0.349
Density of duck farms (n/km^2^)	0.237	0.177, 0.306	<0.001
Density of chicken farms (n/km^2^)	−0.146	−0.300, −0.003	0.048
Human population density (inhabitants/km^2^/100)	0.001	−0.016, 0.017	0.885
Minimum distance to driveway from a farm (km)	−0.156	−0.479, 0.174	0.359

The basic reproduction number was estimated for the HPAI IP genotype in two clusters (42 IP in cluster A and 48 in cluster B). A mixed linear regression model was built including five fixed effects. The cluster was a random effect (intercept). CI, confidence interval. R2 value of the regression model = 82.8%.

## Data Availability

The HPAI H5N6 outbreak data can be found here: https://ebook.qia.go.kr/home/view.php?host=main&site=20180221_093341&listPageNow=0&list2PageNow=0&code=0&code2=0&code3=0&optionlisttype=&searchcode=0&searchcode2=0&searchdate=0&searchkey=allsite&searchval=%C1%B6%B7%F9%C0%CE%C7%C3%B7%E7%BF%A3%C0%DA&searchandor=&dummy=&&orders=A (accessed on 20 May 2019), and non-infected poultry holdings data can be found here: https://www.localdata.go.kr/devcenter/dataDown.do?menuNo=20001 (accessed on 28 July 2016).

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
