# Peer review of "Elucidating the Local Transmission Dynamics of Highly Pathogenic Avian Influenza H5N6 in the Republic of Korea by Integrating Phylogenetic Information"

_pathogens, 2021, doi:10.3390/pathogens10060691_

Round 1

Reviewer 1 Report

Dear Authors,

This is a solid and novel study depicting the need for combining epidemiological analysis with data regarding molecular epidemiology of pathogens. Molecular epidemiology data alllowed the recognition of clusters of specific pathogen strain transmission that makes it possible to have specific preventive measures implemented, that will also be adjusted to the specific spatiotemporal characteristics of the area.

Figure 2: Please refer to Cluster A and B in the Figure legend as you do on the figure (and not Cluster 1 and 2)

Figure 4: Please correct the legend, referring to cluster A and cluster B

Table S3: In the third column, is it No. PCs or No. IPs?

Line 417-419: In table S3 i see a reduction of only 1 IP in cluster A and increase of 31 to No. PCs. Why are you referring to reduction of 2 IPS and increase of 10 PCs here?

Line 420-421: I would suggest the addition of a comment regarding the reduction of the costs that would be done in cluster A if culling range was reduced to 0,5 km rather than 3. It would demonstrate the direct economic advantages implementing this approach on adjusting culling ranges

Reviewer 2 Report

This study by Yoo et al investigates the spatial transmission risk of highly pathogenic avian influenza H5N6 in the Republic of Korea (ROK) after incorporating phylogenetic information into the analyses. The data were obtained from the 2016-2017 HPAI outbreak in ROK and spatiotemporal clusters of infection were identified but were restricted to clusters based on the same genetic variant of virus, providing additional insights into transmission between premises.

Combining genomic information with the spatiotemporal data, two distinct clusters of infection were identified during the 2016-2017 ROK HPAI outbreak. Transmission risk for each cluster was modelled using a transmission kernel function and showed that cluster A had lower secondary attack rates than cluster B. Based on the transmission parameters derived from the kernel function, the authors modelled the number of infected premises vs the number of pre-emptively culled premises in each cluster using 4 different culling radii. This, perhaps unsurprisingly, showed that a larger culling radius resulted in fewer infected premises but more pre-emptively culled holdings.

The authors then go on to claim that the higher density of duck holdings in cluster B led to the increased transmission risk seen in this cluster. However, in my opinion this is completely speculative with no data to support this conclusion. The density of chicken farms was also higher in cluster B and other factors were not considered. There is no analysis to determine whether the duck density was a confounder in this study.

Generally, this is a mostly well-written manuscript and the incorporation of genomic data into the epidemiological clustering is a considerable advancement on previous epidemiological analyses of this outbreak. However, the conclusion that duck farms are responsible for driving transmission risk is unfounded and the choice of only 4 culling radii with no statistical or cost-benefit comparison between them makes it difficult to interpret the findings. There is no measure or evaluation of the ‘optimal’ culling radius. I have listed my concerns below.

Major comments:

Duck farm issue

  • Lines 26-28: Cluster B had both a higher density of duck farms but also chicken farms. Higher transmission risk was not associated only with higher duck-farm density, and this conclusion is inappropriate from the data presented. This should be reframed.
  • Line 361-365 and table 4: these are results, not discussion. Cluster B also has higher chicken farm density. I agree that duck farm density may be a relevant factor, but it may also simply be a confounder. This should be discussed more circumspectly.
  • Line 374-378: from table 4 the chicken farm density was also higher in cluster B. If the argument is a higher proportion of chicken farms compared to duck farms this information should be provided in table 4. But I would still argue that density is more important than overall proportion.
  • Line 411: This study did not evaluate duck vs chicken holdings systematically. The reference to duck farms should be changed to poultry farms more broadly.
  • Line 444: A proper analysis of farm species was not done in this study.

Some sort of formal evaluation of the different culling radii should be performed.

  • Line 416: It is inappropriate to make this statement. Figure 4 shows that 3km did reduce the number of IPs. No statistical analysis was provided for figure 4 so we can’t evaluate whether the reduction was significant or not.
  • Section 3.4 is difficult to follow. The graphs in Figure 4 don’t seem to match with the information provided in Table S3. The values in the text also don’t match with table S3. It is unclear where these numbers come from and what they mean.

Supplementary figures are not cited in the text. It is unclear where the supplementary figures fit and what value they add.  

Minor comments:

Line 191 : Should this be table S1, not fig S2?

Figure 1 is cut off on the right.

Line 237 is missing a full stop.

Fig 2: Can the authors provide any insights as to why a C2 cluster wasn’t identified? Visually it looks like a clear cluster but it would be interesting to understand why the multinomial scan statistics didn’t classify this as a cluster. Was this borderline significant? Or spread over too much time? Or too much distance?

Line 266: Formatting issue – tables 2 and 3 and figure 1 should be placed at the end of this paragraph.

Line 286: should this be ‘no obvious difference’?

Fig S1 legend: C-2 hyphen is in the wrong spot?

Lines 308-309: This sentence is difficult to understand and should be rewritten.

Lines 302-307: Should this be ‘from simulation of’? And ‘culling ranges’? Are cluster 1 and 2 same as cluster A and B? Please use one nomenclature throughout the manuscript for consistency. What units is the x time axis in? What is cluster 3?

Line 316: It is not clear where the 0.5 to 2.5 values come from, or the 23.5 IPs. The graph scale for cluster A goes from 4-~18.

Table S3: should column 3 be IPs?

Line 324: There are visible differences on the graph though. Please provide additional support for this statement.

Line 362: Please use consistent cluster nomenclature throughout the manuscript.

Line 364: Should this be ‘high density duck farms’?

Line 382: Case fatality rate is not synonymous with dropped egg rate. It is unclear what is meant here.

Reviewer 3 Report

General Comments

The authors provide an important and unique study on the impacts of highly pathogenic avian influenza (HPAI) and the spatial dynamics of disease spread among poultry farms in the Republic of Korea. My major concerns raised are in the organization and elaboration of the methods and results.

Detailed Comments

  1. There is a large blank between lines 109-110.
  2. It would be beneficial if the authors could provide a definition of how they defined an HPAI outbreak.
  3. It would be helpful if the authors wrote out full years throughout the text. In line 112: “2016 to 2017” instead of “2016-17.”
  4. The methods could read better if the first section included background information on the study system. If the authors could provide geographic regional context of the selection of the Korean farms that could be useful.
  5. The methods used for phylogenetic analysis would read better as its own paragraph.
  6. Lines 125-126 are speculation. This needs to be omitted from the methods.
  7. It appears the authors selected a multinomial distribution because of the categorial classification of their variables. The text would read better if the authors could better clarify their outcome and predictor variables and choice of distribution.
  8. The authors should provide justification for the choice of 20% for their scanning window. Typically, 50% is the acceptable minimal value.
  9. Lines 169-171 need to be introduced earlier in the study system. The methods could be enhanced if the authors could include all the sampling and experimental design in the first paragraph. Some of this essential content is in the text but difficult to assess (e.g. localities, time periods, geographic regions, poultry breeds, number of farms, flock sizes (range/mean).
  10. It would also be useful if the authors could elaborate on the time period of sampling and cluster analysis. It appears these are season-based, providing some more context would be very helpful to the audience.
  11. Could the authors elaborate on the radius used for their simulated ring culling?
  12. Lines 263-266 are incomplete. It is unclear how the authors are extrapolating the kernel density proportions into a model comparison between areas.
  13. Figure 3 is very informative. It looks like this was factored in the simulated radius measurements? If so, I suggest the authors reorder the results in reporting the results from Fig. 3 earlier.
  14. Some of the figure images are a bit grainy. For the final submission, it would enhance the quality of the manuscript if the authors submit 300 DPI images.

Round 2

Reviewer 2 Report

The authors have addressed many of my initial concerns. Thanks for their prompt work addressing these changes. I am confident that the appropriate analyses have now been done to support the assertion that duck farms are the primary drivers of transmission, based on this analysis.

In response to the changes to the manuscript, I have some additional minor corrections. Additionally, the revisions need extensive English language editing.

Line 113: This should be RT-qPCR.

Line 164: Should this be ‘expected number of cases’?

Section 2.4: R0 refers to the average number of secondary infections per case in a fully susceptible population and is a fundamental parameter of the virus that is really more of a theoretical concept. Since you are calculating an actual value for individual IPs, is it more appropriate to call this the secondary attack rate?

Table 4: p values or odds ratios should be given for the different variables.

Lines 393-394: The authors explain that the number of IPs doesn’t vary much with different culling radii in Cluster A (17-18), according to the simulation model. However, the observed number of IPs was much higher than the simulation model predicted (42 vs 18). Doesn’t that suggest that the simulation model is not appropriate and needs refining? Can the results from this simulation model, which clearly does not reflect the real-world scenario, realistically be used to guide meaning results?

Reviewer 3 Report

Thank you to the authors for taking the time to address all of my comments! The paper is a very strong product!

Author Response

Dear Reviewer, 

Thank you for your time in reviewing my research work. Your comments make this paper well-organized and appealing to readers.